# A refinement approach in a mouse model of rehabilitation research. Analgesia strategy, reduction approach and infrared thermography in spinal cord injury

Veronica Redaelli[1]☯, Simonetta Papa[2]☯, Gerardo Marsella[3], Giuliano Grignaschi[3], Alice Bosi[4], Nicola Ludwig[5], Fabio Luzi[1], Irma Vismara[2], Stefano Rimondo[2], Pietro Veglianese[2], Svetlana Tepteva[1], Silvia Mazzola[1], Pietro Zerbi[6], Luca Porcu[7], John V. Roughan[8], Gianfranco Parati[4,9], Laura Calvillo[4]*

1 Dipartimento di Medicina Veterinaria, Università degli Studi di Milano, Milan, Italy, 2 Department of Neuroscience. Laboratory of Biology of Neurodegenerative Disorders, Istituto di Ricerche Farmacologiche Mario Negri IRCCS, Milan, Italy, 3 Animal Care Unit, Istituto di Ricerche Farmacologiche Mario Negri IRCCS, Milan, Italy, 4 Department of Cardiovascular, Neural and Metabolic Sciences, S.Luca Hospital, Istituto Auxologico Italiano IRCCS, Milan, Italy, 5 Dipartimento di Fisica, Università degli Studi di Milano, Milan, Italy, 6 Dipartimento di Scienze Biomediche e Cliniche "L. Sacco", Università degli Studi di Milano, Milan, Italy, 7 Department of Oncology, Laboratory of Methodology for Clinical Research, Istituto di Ricerche Farmacologiche Mario Negri IRCCS, Milan, Italy, 8 Institute of Neuroscience, Comparative Biology Centre, Newcastle University, Newcastle upon Tyne, United Kingdom, 9 Department of Medicine and Surgery, University of Milano-Bicocca, Milan, Italy

☯ These authors contributed equally to this work.
* l.calvillo@auxologico.it

**Data Availability Statement:** All relevant data are within the paper and its Supporting Information files.

## Abstract

The principles of Refinement, Replacement and Reduction (3R's) should be taken into account when animals must be used for scientific purpose. Here, a Reduction / Refinement approach was applied to the procedure of spinal cord injury (SCI), an animal model used in rehabilitation medicine research, in order to improve the quality of experiments, avoiding unnecessary suffering. The aims of this investigation were 1- to assess acute surgical pain in mice subjected to SCI, 2- to compare the efficacy of commonly used analgesia (three buprenorphine subcutaneous injection in 48 hours, 0,15 mg/kg each) with a combination of opioid and NSAID (one subcutaneous injection of 5 mg/kg carprofen before surgery followed by three buprenorphine subcutaneous injection in 48 hours, 0,15 mg/kg each) and 3- to test if Infrared Thermography (IRT) could be a potential new Refinement method to easily assess thermoregulation, an important metabolic parameter. Finally, we aimed to achieve these goals without recruiting animals on purpose, but using mice already scheduled for studies on SCI. By using behaviours analysis, we found that, despite being commonly used, buprenorphine does not completely relieve acute surgical pain, whereas the combination of buprenorphine and carprofen significantly decreases pain signs by 80%. IRT technology turned out to be a very useful Refinement tool being a non invasive methods to measure animal temperature, particularly useful when rectal probe cannot be used, as in the case of SCI. We could find that temperatures constantly and significantly increased until 7 days after surgery and then slowly decreased and, finally, we could observe that in the

**Funding:** The authors received no specific funding for this work.

**Competing interests:** The authors have declared that no competing interests exist.

buprenorphine and carprofen treated group, temperatures were statistically lower than in the buprenorphine-alone treated mice. To our knowledge this is the first work providing an analgesic Refinement and a description of thermoregulatory response using the IRT technology, in mice subjected to SCI.

## Introduction

According to European directive 2010/63 on the protection of animals used for scientific research, the principles of Refinement, Replacement and Reduction (3R's) have to be taken into account when animal models must be used for scientific purpose. Our previous work [1] described an ethical approach to set up the analgesic management of thoracic post-surgical acute pain, that is quantifying and managing pain in animals already scheduled for biomedical research experiments, without recruiting animals on purpose. This approach allowed to reach two important goals, first it was possible to Reduce the number of animals used and, second, allowed researchers to Refine their surgical model by observing and quantifying stress/pain related behaviours and by developing more accurate analgesia.

In connection with that experience, our group applied the same Reduction approach to refine experimental surgical procedure of spinal cord injury (SCI), an animal model used in rehabilitation medicine research.

According to the World Health Organization at least 250,000 people suffer a SCI each year due to road traffic crashes, falls or violence, becoming disabled, and these patients are two to five times more likely to experience comorbid health problems leading to premature death [2]. Both the societal cost of SCI and the individual patient costs are vast, consequently to refine the experimental model of SCI is extremely important for the preclinical research aimed to develop potential treatment and rehabilitation strategies. Medical research needs solid experimental models and applying 3R's, with the development of new methods to Refine and Reduce, can improve the standard of animal models used in rehabilitation medicine.

SCI could potentially be severely painful to mice, analgesics are therefore essential to avoid unnecessary suffering, low quality data and waste of animal and financial resources. Unalleviated pain may interfere with study integrity since pain and stress can result in major physiological alterations and statistical noise [3–4]. Another major issue is that despite substantial research efforts there is still only limited information about the analgesic needs of mice. At present, little is known about pain severity and its prevention in mice used as models of SCI. Despite guidelines [5] recommend association between opioid and NSAID in mice surgery, the best match results in pubmed research, for the keywords "spinal cord injury mouse", report articles in which analgesia is not mentioned or it is limited to Buprenorphine [6–10]. The use of drug combination is very common and already studied in various kind of animal species including primates and also in various pain models, nevertheless, in the mouse model of SCI this approach is not yet commonly applied. The aims of this investigation were therefore the following: a) to assess acute surgical pain in mice subjected to SCI, b) to compare the efficacy of commonly used analgesic strategy (buprenorphine injection) with a multimodal approach (carprofen injection followed by buprenorphine injections) and c) to test if Infrared Thermography (IRT) could be a potential new Refinement method. IRT is a non-invasive technique that has been found to reflect welfare, physiologic and metabolic status in several species [11–15], whereby the dynamics of peripheral blood flow is altered in stressful or fearful conditions as well as in those resulting in pain [16–17]. IRT has proven to be a potential tool for

Refinement, assessing superficial temperature, at distance, in a completely non-invasively way [18]. Moreover, it was yet used in rat and rabbit to study reactions after stress stimuli [19] [17], because stress, fear and pain, can create vasoconstriction/vasodilation phenomena altering blood flow. Despite this, to our knowledge there are no works indicating the reference temperatures for mice undergoing surgery events. Finally, according to its Refinement aspirations, the study was conducted in line with the 3R's principle of Reduction; i.e. the assessments were conducted in mice that were already scheduled for SCI without recruiting animals on purpose.

## Methods

### Ethical statement

The protocol was approved by the Institutional Animal Care and Use Committee or Animal Welfare Body (AWB) of the Istituto di Ricerche Farmacologiche Mario Negri IRCCS (Milan Italy) and the study was carried out in strict accordance with the recommendations and with the authorization of the Italian Ministry of Health (Permit Number 62/2014-PR) according to 26/2014 Italian Law on the protection of animals used for scientific purposes. All surgery was performed under isoflurane anesthesia, and all efforts were made to minimize suffering. The manuscript was prepared according to the ARRIVE guidelines [20–21].

### Refinement and reduction

In compliance with the 3R's principle of Reduction, animals included for the present analysis were selected from those already scheduled for studies on SCI and the monitoring methods were designed to have no impact on the primary study outcomes.

### Husbandry

Males C57BL/6N mice, average weight 25 gr, about 2 months old, were supplied by Charles River (Calco, LC, Italy), were kept in cages with wood-shaving bedding (each cage of 435 cm2 will house 5 mice) and permitted one week to acclimate to their new surroundings prior to experimentation. Male gender was chosen to avoid possible hormonal interferences, thus preventing potential bias in experimental results. All cages were open to the room environment and animals were regularly handled by the staff involved in the experimental protocol. Room temperature were maintained at 20˚±2˚C with 55% ± 10% relative humidity and ventilated at 15–18 filtered air changes per hour. Animals were kept on a 12:12 light: dark cycle and provided ad libitum access to water and rodent feed (Teklad Global Diet 18%—ENVIGO RMS). Considering the type of surgical operation which the animals were subjected to, and the position of the wound, the designated veterinarian chose absorbent paper as a nesting material. Two mice for cage were housed after surgery. Regular blood screenings on sentinel mice certified the absence of endoparasites and ectoparasites in the animal facility which is classified as SPF (specific pathogen free).

### Study design

To assess pain and stress in mice subjected to SCI, to compare the efficacy of two different analgesic strategies and to test IRT as new Refinement method, the study was performed in two steps, each lasting 28 days (Fig 1a and 1b).

STEP-1. assessment of acute surgical pain, stress and IRT in mice after SCI, under commonly used analgesic strategy, consisting in injection of 0,15 mg/kg buprenorphine (BUP) subcutaneously (s.c.) three times in 48 hours. First injection was performed 15 minutes before surgery, the second and the third, 24 and 48 hours respectively after surgery (Fig 1a).

**A**

*STEP-1: Systematic behavior analysis to identify and quantify specific pain and stress in SCI procedure*
*28 days study, **Standard Analgesic Treatment (0,15 mg/kg BUP s.c.); n=12***

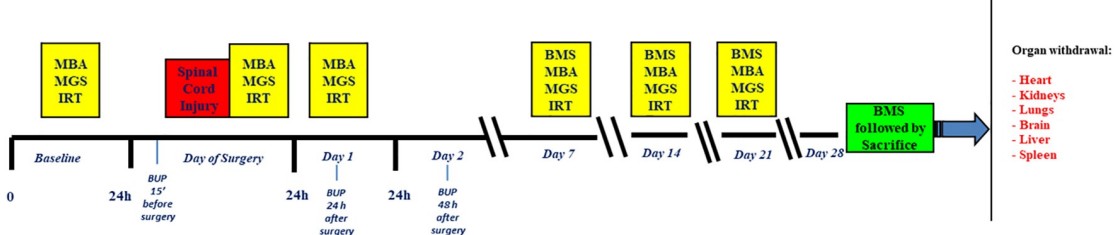

**B**

*STEP-2: Comparison between BUP and BUP+CAR analgesic treatment in SCI procedure*
*28 days study, **Standard Analgesic Treatment (0,15 mg/kg BUP s.c.) Controls n=12***
*vs **Multimodal Analgesic Treatment (0,15 mg/kg BUP+ 5 mg/kg CAR s.c.) Treated n=12***

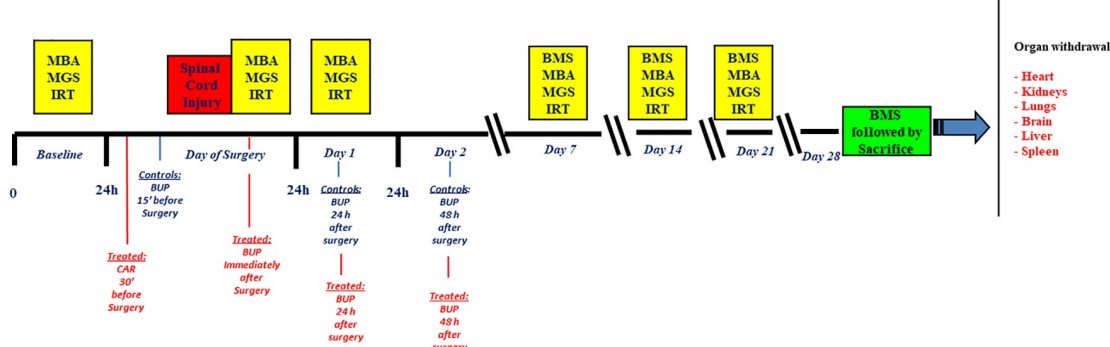

**Fig 1. Study design.** The study was conducted in two steps; in STEP-1(A) only BUP was provided, 15 minutes before SCI, and then repeating treatment at 24 and 48h. The MBA, IRT and MGS assessments were applied before surgery (Baseline, as a non-surgery control), immediately post-surgery (Day 0), and then at 24h (Day 1), and 7, 14 and 21 days. In STEP-2 (B), a control arm, identical to STEP-1, was compared to an experimental arm consisting in a multimodal analgesic strategy (BUP+CAR). Both steps were followed by retrieval of various organs for histological evaluations.

**STEP-2**. comparison of two analgesic strategies: assessment of acute surgical pain, stress and IRT measurements after SCI under a) BUP analgesia (same BUP injections as in STEP-1) or b)under a combination of two analgesic drugs [multimodal strategy [22]]: one s.c. injection of 5 mg/kg carprofen (CAR) 30 minutes before surgery, followed by one s.c. injection of 0,15 mg/kg BUP immediately after surgery, and two further BUP s.c. 24 and 48 hours respectively after surgery (BUP+CAR analgesia) (Fig 1b).

We chose the multimodal strategy after our experience with thoracic surgery analgesia [1] and the type of drugs and doses on the base of the literature [22] and of our experience. In fact, in our hand the commonly used BUP injection every 12 hours [5] caused kidney failure in most SCI mice, likely due to a decreased blood perfusion in this organ, already affected by the hind limbs paralysis. We therefore decided to reduce the frequency of opioid injections.

The following procedures were performed to evaluate post-surgical behaviour, acute surgical pain, stress and IRT: Basso Mouse Scale (BMS) [23], Manual Behaviour Analysis (MBA) [24], Mouse Grimace Scale (MGS) [25] and temperature detection in the brown adipose tissue (BAT) area with a high resolution infrared camera [16–17]. These procedures consist of a

series of video recordings to detect predefined signs of pain and/or stress and BAT temperature [1] [24–25]. MBA, MGS and IRT were performed during the first week at baseline (24 hours prior to surgery), at day 0 (immediately after recovery from surgery), and again after 24 hours (day 1). After the first week, the measurements were recorded at day 7, day 14 and day 21. BMS was performed at day 7, day 14, day 21 and day 28 (Fig 1a and 1b).

There was only one experimental group in STEP-1 (BUP treated, n = 12 mice) and two experimental groups in STEP-2:

- Control (BUP), (n = 12)

- Treated (BUP+CAR), (n = 12)

Pain assessment was evaluated through the observation of video-recordings, which occurred a posteriori in single-blind conditions, by two investigators. Animals were randomly assigned in a 1:1 ratio by using a fair coin.

## Experimental procedures

**Surgery.** All surgical equipment was provided by 2Biological Instruments, VA, Italy. Drugs were provided by veterinary pharmacies: Isofluorane-Vet isofluorane, Merial, Italy) Temgesic (buprenorphine, Indivior, UK) Imalgene (ketamine hydrochloride, Merial, Italy) Domitor® (medetomidine, Vétoquino, Finland) Rimadyl® (carprofen, Pfizer, Italy) and Amplital® (ampicillin, Pfizer, Italy).

**Pre-surgical preparation**: Each researcher wore disposable clothes, gloves, footwear, surgical mask and a hair cover.

The operating table and heating pad were disinfected with TEGO 51, an aqueous mixture of amphoteric amines. The mouse skin was disinfected with chlorhexidine gluconate 4%, before surgery. The surgical tools were autoclaved.

Ampicillin 50 mg/kg was injected s.c. 5 minutes before surgery and for the subsequent 4 days, as modified protocol based on our previous study [1].

**Surgery**: *Anaesthesia and SCI* Before surgery, animals received one s.c. injection of ampicillin (50 mg/kg) and analgesic treatment as described above and in Fig 1 (BUP or BUP+CAR). Surgical procedure was carried out in deep anaesthesia by isoflurane (3% in $O_2$). After hair removal and disinfection, animals were placed on a Cunningham Spinal Cord Adaptor (Stoelting, Dublin, Ireland) and laminectomy of T12 vertebra was done to uncover the lumbar spinal cord. Trauma was induced using an aneurysm clip with a closing force of 60g (left in place for 1 min and then removed).

At the end of the procedure, animals underwent surgical suturing and treatments until complete recovery (body temperature control with heating pad until complete awakening after anaesthesia and 50 mg/kg of ampicillin plus rehydration every day for 4 days).

**Spinal cord surgery-related behavioural evaluations: Locomotor performances analysis (BMS).** All treated mice were evaluated by testing hind limb locomotor performances using BMS once a week from day 7 to 28 Days Post Injury (DPI) (when animals started to recover partial mobility). The BMS is a 10-points scale (9 = normal locomotion; 0 = complete hind limb paralysis). Video acquisition of the locomotor performances (5 min) was performed by a ICD-49E camera (Ikegami) and the evaluation was carried out by two independent observers, blinded to the treatment. Individual hind limb scores were averaged for each animal group at each time point.

**Acute surgical pain and wellness assessment: MBA, MGS, IRT.** The cameras to assess MBA, MGS and IRT were two Sony Digital Handycam and a microbolometric high resolution infrared camera FLIR with A65, 640X512 pixel sensor.

Each recording session lasted about 30 minutes (3 minutes for the first IRT measurement, two consecutive videos, lasting 10 minutes each, for MBA and for MGS and again one IRT measurement lasting 3 minutes).

IRT was performed before any other manipulation, when the mouse arrived from the housing area (T0), and 30 minutes after (T30), when mouse completed the behavioural surveys exploring different environments. The fur in the region of interest was cut the day of the surgery and temperatures were recorded always between 10 a.m. and 1 p.m.

Three IRT images for each subject were stored and evaluated with Flir Tools and IRT Analyser software and the median of the max temperature values on the BAT and on the tail were extracted. To analyse thermoregulatory behaviour, only T30 values were used, in order to avoid possible confounding effect of anaesthesia at T0 in the day of surgery.

To assess behaviours indicating acute surgical pain and stress, each mouse was filmed in polycarbonate cages for analysis of predefined pain and stress specific indicators [24–25]. Pain severity was assessed via MBA scoring, encompassing the presence and the frequency of pain and stress specific signs [24].

According to Langford [25], the MGS was applied by evaluating the ear, eye, cheek, nose bulge and whisker Facial Action Units (FAU's) [25]. An MGS score between 0 to 2 was assigned to quantify pain (whereby a score of 0 represents pain not present, 1 represents moderate pain and 2 represents severe pain).

Considering that the procedure of video recording does not cause any stress or suffering, we recorded all the animals used in the main study to have the maximum numerousness possible.

All assessments were performed by two observers, blind to the treatment, for a single blind control.

**Euthanasia.** For the main study-primary endpoints, animals were sacrificed according to the authorized method of transcardial perfusion after overdose of ketamine hydrochloride (150 mg/kg) and medetomidine (2 mg/kg) anaesthesia and were perfused with 40 ml of PBS, 0.1 mol / liter, pH 7.4, followed by 50 mL sodium 4% paraformaldehyde solution buffered with phosphate.

Organs were then collected and stored in formalin for histological analysis.

**Histology.** Histological analyses were performed in order to verify the status of the organs after surgery and the consequent paralysis of the hind limbs. Histological analyses allowed to eventually exclude those animals showing abnormal features not related to the procedure, thus verifying the quality of surgical procedure itself. The organs included in paraffin embedded blocks were liver, heart, lungs, kidneys, brain and spleen.

After fixation, sampling was performed in order to obtain representative paraffin embedded blocks of each organ. Three micrometer (3 μm) thin sections were cut and H&E staining was performed for all blocks. Trichromic Masson's stain was performed on organ samples in order to evaluate tissue fibrosis. Digital images were obtained by acquiring slides with a digital scanner (Hamamatsu NanoZoomer-SQ, Japan). All organs were analysed in order to detect any histological abnormality, architectural or inflammatory.

**Statistical analysis.** The number of animals to be included was derived from power calculation for the major endpoint of the main study, which was the improvement of locomotor functions (BMS) of injured mice after different neurodegenerative treatments. In particular, a sample size of 12 animals per group is required to detect a difference of 3 in BMS with a standard deviation (SD) of 3.3 and a correlation of 0.5 between matched pairs, with a significance level of 0.05 and a power of 0.80.

A non-parametric approach was used to statistically describe and formally test animal data. Specifically, covariate distributions, treatment characteristics and outcomes were summarized

using non-parametric descriptive statistics (median and range for ordinal and continuous variables, and absolute and percentage frequencies for categorical variables). In case of respectively paired and unpaired data, Wilcoxon signed-rank test and Wilcoxon rank-sum test were used to formally detect difference between groups. The Friedman test was used to formally detect differences between groups across repeated measures. Statistical analysis was generated using SAS/STAT software, version 9.4 of the SAS system for Windows. Range plots with capped spikes were generated using Stata Software, version 15.1 (StataCorp. 2017. College Station, TX: StataCorp LLC).

## Results

Complete raw data are available in Supporting Information files: S1 and S2 Files.

### Spinal cord surgery-related behavioural evaluations

**Locomotor performances analysis (BMS).** **STEP-1**: Twelve mice underwent SCI and one was found dead 48 hours after surgery. A physiological recovery up to median 2.0 BMS score at 28 DPI (range: 2–3) was detected after an initial complete paraplegia corresponding to a BMS score of 0, which was verified for all animals included in the study (Fig 2A).

**STEP-2**: Twenty-four mice (twelve per group) underwent SCI. Three mice that received BUP alone were euthanized on welfare grounds; the first at 24 hours after surgery, another after 48 hours and the third on day 14. One BUP+CAR treated mouse completed 21 days of behavioural testing but was found dead on day 26. BMS score confirmed the physiological recovery up to median 2.0 BMS score at 28 DPI (range: 2–3) in both mice treated with BUP and BUP+CAR (median: 2.0; range: 1–3). No statistically significant difference was detected between arms (p-value, Friedman test: 0.757), suggesting that the alternative analgesic protocol does not affect the quality of the experimental procedure (Fig 2B).

### Manual Behaviour Analysis (MBA)

In both STEP 1 and 2, the only sign detectable among behaviours associated with suffering [24] was twitching (an intense short-lasting muscle spasm), no abdominal writhing, nor backarching or staggering were observable. Among the typical rodent behaviours, only grooming (a series of behaviours like paw licking followed by face washing, head scratching, etc), digging and normal cage exploration were present. It should be noted that SCI procedure completely paralyzes hind limbs, therefore many behaviours, including some indicating distress, were impossible to perform by the animals.

**STEP-1**. Twitching behaviour was not apparent before SCI but significantly increased relative to baseline during the first 24 hours after surgery. The median scores on the day of surgery and 24 hours later were, respectively, 8.0 events/10 minutes (range: 4–19, p = 0.001) and 1.0 events/10 minutes (range: 0–3, p = 0.008) (Fig 2C).

Twitching seems to be specifically associated with acute pain in the thoracic area, both ventral and dorsal, in small laboratory rodents. This behaviour has been observed in rats [1] after thoracotomy and in mice during this SCI study. Moreover, it was observed that signs of wellbeing such as grooming, digging and normal cage exploration, found in healthy animals, were greatly reduced during the 24 hours post-operation and they resumed normal frequencies in the following weeks.

**STEP-2**. Overall, multimodal analgesia (BUP+CAR) was significantly more effective than BUP alone in reducing twitching frequency (Friedman test; p<0.001), especially immediately post-surgery [median BUP 7.5 events/10 minutes, (range: 0–23) vs. BUP+CAR 1.0 events/10

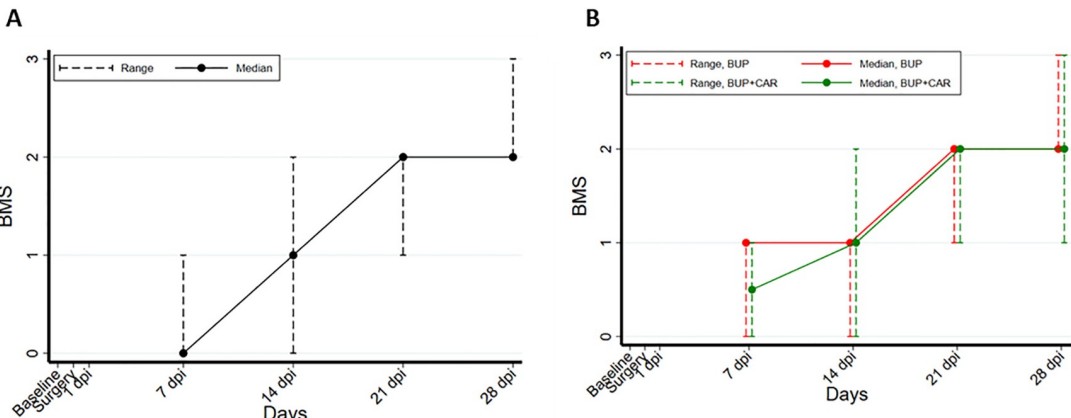

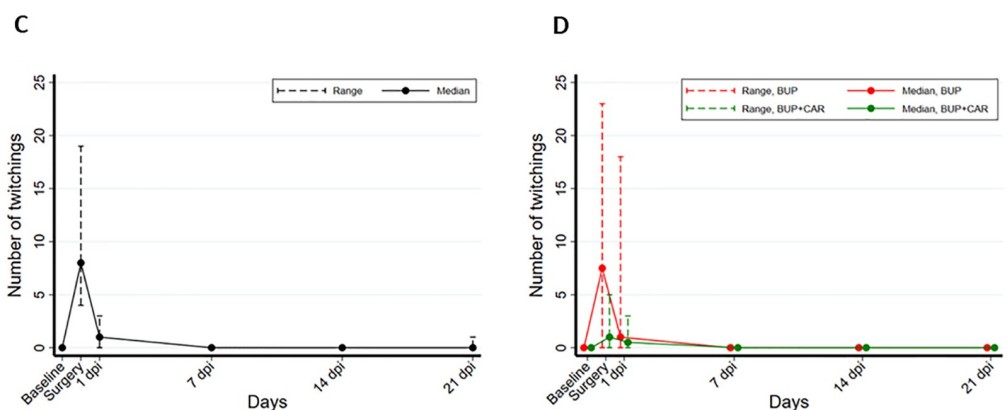

**Fig 2.** Locomotor performance (BMS) results in mice undergoing SCI in STEP-1 (A) and STEP-2 experiments (B). Assessments were undertaken at 7, 14, 21 and 28 DPI. Time course of twitching activity in STEP-1 (C) and STEP-2 (D) experiments. This sign of pain increased during the day of surgery in both settings, but in BUP treated animals of STEP-2 experiment, it statistically increased with respect to BUP+CAR group (p = 0.001).

minutes, (range: 0–5; p = 0.001)], thus suggesting that combination of BUP+CAR in this surgical model is more effective in reducing acute pain than only BUP (Fig 2D).

As in STEP-1, also grooming and digging were reduced during the first 24 hours after surgery, but not significant difference was found between the two groups of analgesic treatments. Mice thereafter regained and maintained normal digging and grooming.

**Mouse Grimace Scale (MGS).** **STEP-1**: Among the MGS components analysed (eyes, cheeks, whiskers and nose), the most indicative of suffering was the ear position, as seen in other species like laboratory rabbit [26]. On the base of the ears position, pain is reflected by a pulling apart and back of the ears from baseline position. Before surgery, animals showed baseline ear position with a pain score of 0, but during the first 24 hours after surgery the ears became flattened to the skull, with the apex turned caudally [median: 1.25 (range: 0.5–2, p = 0.001) the day of surgery, median: 1.5 (range: 0.5–2, p = 0.001) the day after surgery]. Over

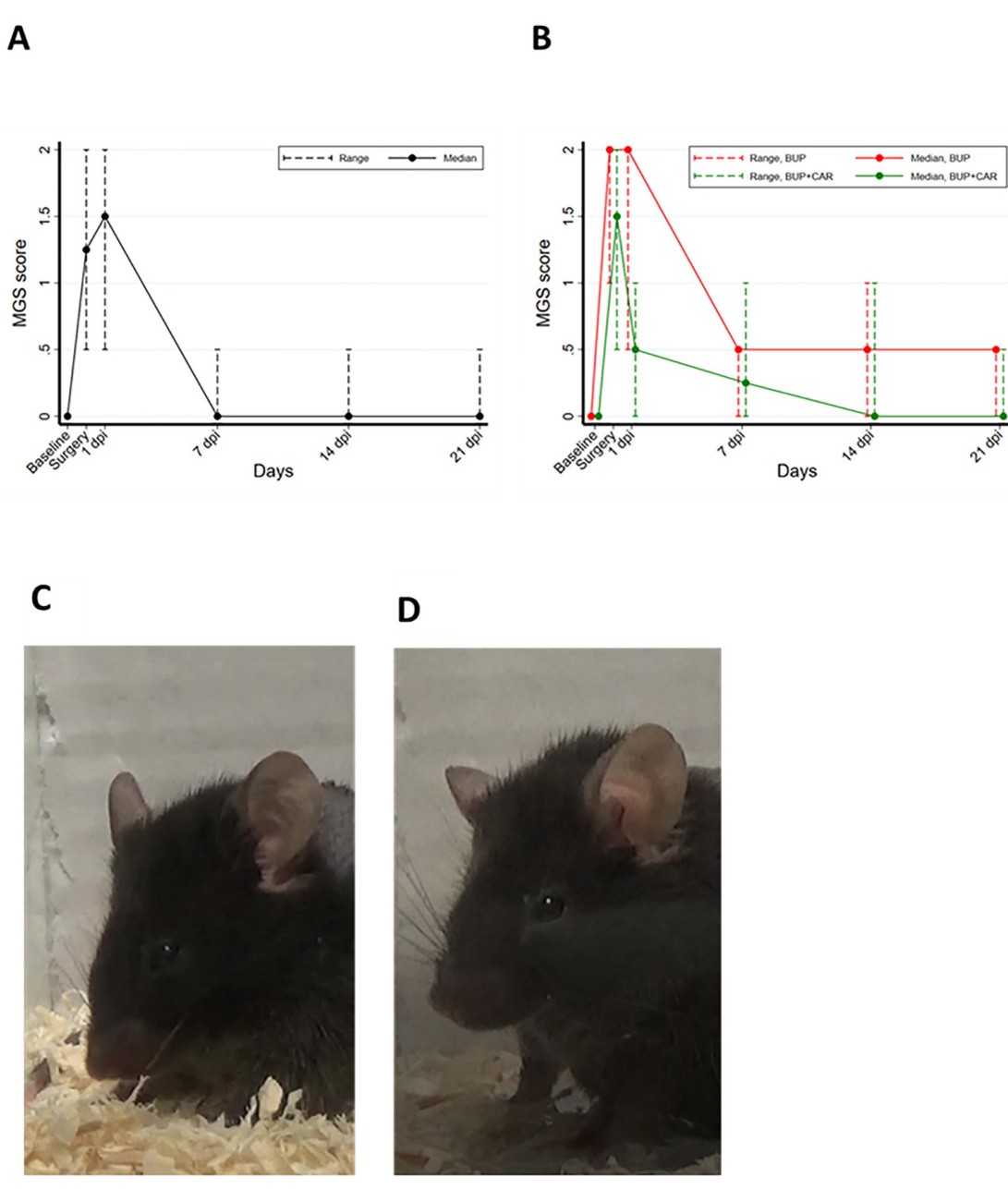

**Fig 3. Pain score based on ear position according to MGS method.** A: STEP-1, score increased during the day of surgery until 24 hours later and decreased to 0 one week later. B: STEP-2, BUP+CAR treatment statistically decreased pain score 24 hours after surgery. As in STEP-1, score returned to baseline value after one week. Example of ear position in the mouse C: BUP treated animals 24 hours after surgery, D: BUP+CAR treated animals 24 hours after surgery.

the subsequent weeks the ears resumed a partially normal position but never fully recovered (i.e. completely erect) (Fig 3A).

STEP-2: Overall, BUP+CAR treatment was more effective than BUP alone in reducing pain according to ear position score (p-value, Friedman test: <0.001). There was a significant difference between BUP and BUP+CAR groups on both the day of surgery [median: 2.0 (range: 1–2) in the BUP group vs median: 1.5 (range: 0.5–2) in BUP+CAR group, p = 0.021)], and on the day after surgery; in this latter case, mice in the BUP+CAR group showed a more rapid

recovery respect to BUP treated animals, [median: 2.0 (range: 0.5–2) in BUP group vs median: 0.5 (range: 0–1) in BUP+CAR group, p = 0.002] (Fig 3B, 3C and 3D).

**IRT.** Although IRT recordings were performed on both the tail and BAT, the tail data were not included as they were found to be highly variable. This was possibly a metabolic consequence of SCI; affecting the entire caudal section of the mouse. By contrast, results from the BAT area were consistent and reproducible and allowed us to describe the thermoregulatory response to surgery.

**STEP-1**: Baseline temperatures in the brown adipose tissue (BAT) area were lower than subsequently recorded. The difference was statistically significant at 24 hours after surgery both respect to baseline [median: 32.6˚C 24 hours after surgery (range:31.3–34.3) vs. 30.8˚C (range: 29.4-31-4) in the baseline, p = 0.001] and to the day of surgery [median: 32.6˚C (range:31.3–34.3) 24 hours after surgery vs median: 31.2˚C (range: 29.2–32.6) in the day of surgery, p = 0.005]. Temperatures constantly increased until the day 7, and then slowly decreased (Fig 4A).

Low values were documented in the surgery day even if these data were detected with no fur, as the area was shaved immediately before surgery.

**STEP-2**: In the BUP treated animals, we confirmed the temperature trend observed in STEP-1, despite the absolute temperature values were different. In the BUP+CAR treated group, temperatures were systematically and statistically lower than in the BUP group (p-value, Friedman test: 0.011). Maximum difference was observed at day 14 [median: 36.4˚C (range: 33.2–40.4) in BUP+CAR vs median: 39.9˚C (range: 33.1–40.4) in BUP] (Fig 4B).

**Histology.** Histological analysis showed typical alterations of the model but not obvious signs of failure, confirming the correct execution of the surgical procedure. No experimental subject was therefore excluded for abnormality related to histological findings. All organs displayed normal architecture except for some alterations such as vascular damage and hypoxia of the liver, likely due to surgical-induced paralysis of hind limbs. Fibrosis was specifically evaluated using Masson's trichromic stain, but no alterations were found in any treatment group. Presence of inflammatory cells was found in some samples, but none of these results were pathologically significant. Rare lymphocytes infiltrates were dispersed in liver parenchyma without statistically differences between the two groups.

## Discussion

The main goal of this project was to apply a Refinement and Reduction approach to a mouse model of SCI, which is still essential for rehabilitation research, with the aim of improving the standard of animal models used in rehabilitation medicine. This was achieved through three main interventions: the first was to use mice already scheduled for studies on SCI, without recruiting animals on purpose (Reduction). The second intervention was to verify the actual level of stress and suffering experienced by mice undergoing SCI when treated with the commonly used analgesic drug (buprenorphine) and, consequently, to test if the recommended association between opioid and NSAID could improve analgesia after SCI surgical procedure (Refinement). The third intervention was to test a potential new Refinement method (IRT) already used in rat and rabbit but not yet in laboratory mice.

Regarding the first point, that is studying and improving Refinement in animals already scheduled for experimental procedures, we hope that this work will encourage investigators to do likewise, filming behaviours as routine during experiments. This will allow researchers to detect and avoid possible bias caused by pain and stress, thus improving the quality of experimental data.

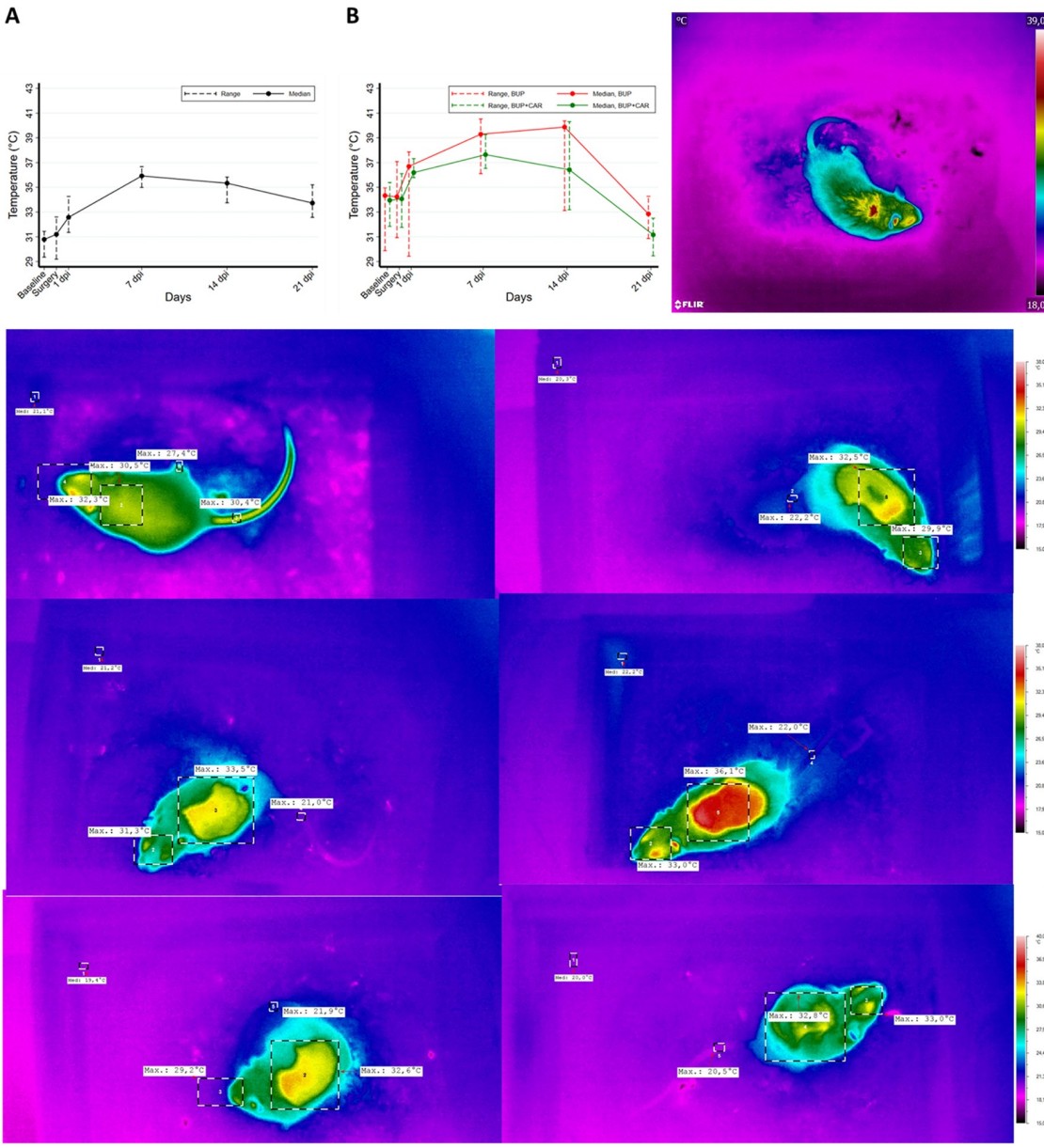

**Fig 4. IRT measurements.** Upper panel left: In STEP-1 (A), at the baseline, the temperatures recorded in the BAT area were lower than those recorded in the day of surgery and in the following days. In the STEP-2 experiments (B), BUP+CAR treated group showed temperatures statistically lower than in the BUP group. Upper panel right: example of infrared images obtained, the hottest region in red is the BAT area of the mouse. Lower panel: sequence of infrared images of a mouse from baseline (upper left) to day 21 (lower right). The highest temperature were recorded on day seven (middle right).

The second issue was pain and analgesia. The problem of pain has a remarkable impact in refining animal models and several studies performed on laboratory animals demonstrated that pain is associated with profound changes in the physiology of the immune and nervous systems and in gene expression [27–29]. At present, buprenorphine is commonly used for experimental SCI procedure, nevertheless observations made in STEP-1 experiments showed unexpected signs of suffering in our animals. We verified that the recommended treatment [5] with a combination of opioids and NSAID, the BUP+CAR treatment, was more effective than

the opioid alone in decreasing surgical acute pain. Therefore, this type of analgesia could be considered for SCI surgery in all those settings where the use of NSAID does not create drug-related confounding effects.

Another important observation was the presence of twitching when there was pain in the thoracic area, a finding already noted in our last work [1], suggesting that the anatomical site of injury might affect different suffering behaviours. There is therefore the need of assessing specific pain and stress signs for every in vivo procedure and there is the need of a precise quantification and management of pain.

Finally, IRT can be considered a new refinement tool being a not invasive methods to measure animal temperature, particularly useful when rectal probe cannot be used, as in the case of SCI.

Thermoregulation is a very important physiological and metabolic parameter [11–13] that should be taken into account in every experimental model. IRT allows a rapid assessment, over only a 3 minute recording period, thus further reducing animal stress. To our knowledge this is the first work providing a description of mice thermoregulatory response after SCI surgery using the IRT technology, and showing a possible correlation between thermography and pain. In fact, we could describe the temperature trend during the first week after surgery and we could observe that BAT temperatures of BUP+CAR treated mice were systematically and statistically lower than in the BUP group. We cannot clarify whether this temperature decrease in BUP+CAR group was due to an anti-inflammatory effect of the NSAID component of the multimodal analgesia or to another reason since, in order not to interfere with the main study procedures, we could not perform biochemical measurements, nevertheless this is an issue which deserves to be deepened in future studies.

## Study limitation

There are some study limitations: the monitoring methods were designed to have no impact on the primary study outcomes, therefore, during the three weeks of the study it was not possible to withdraw blood to measure pain-related peptides levels or inflammatory parameters. Also nesting-score was not possible since the particular type of surgery wound required ready-to-use nest enrichment. Nevertheless, all the observations made in the Refinement study might have an impact also for the primary study, therefore, indirectly, for translational outcome. Moreover, because of the posterior limbs paralysis, we have excluded from the analysis various behaviour associated with pain, such as 'cat like' back arch, writhing and stagger/fall. For the same reason, some welfare signals, typical of well-being or post-operative recovery of activities, such as jumping or rearing, could not be included in our analysis.

## Conclusion

Biomedical research aimed to patient rehabilitation needs solid experimental models. It is urgent to verify and quantify stress and hyperalgesia in laboratory animals since, from their metabolic response in experimental models of human pathologies, we may collect results useful for switching from bench to bedside. In this regard, this work provides interesting information. It offers a description of pain assessment in mice subjected to SCI; it compares two different analgesic strategies, identifying the most effective one and it describes mice thermoregulatory response after SCI surgery using the IRT technology, showing a possible correlation between thermography and pain. Finally, it applies a Reduction strategy to the study design, thus achieving the purpose of applying the Reduction/Refinement principle to an in vivo study.

## Supporting information

**S1 File. Step1 rawData pdf file includes the complete database of the step 1 study (animals treated with BUP analgesia only).**
(PDF)

**S2 File. Step2 rawData pdf file includes the complete database of the step 2 study (animals treated with BUP vs BUP+CAR analgesia).**
(PDF)

## Acknowledgments

Authors are grateful to Dr. Pietro Cipresso, Dr. Juan Eugenio Ochoa and Dr. Lidia Cova for their precious suggestions.

## Author Contributions

**Conceptualization:** Pietro Veglianese, Pietro Zerbi, Laura Calvillo.

**Data curation:** Veronica Redaelli, Simonetta Papa, Alice Bosi, Nicola Ludwig, Pietro Veglianese, Pietro Zerbi, Luca Porcu, John V. Roughan, Laura Calvillo.

**Formal analysis:** Veronica Redaelli, Simonetta Papa, Alice Bosi, Luca Porcu, Laura Calvillo.

**Investigation:** Veronica Redaelli, Simonetta Papa, Gerardo Marsella, Alice Bosi, Fabio Luzi, Irma Vismara, Stefano Rimondo, Svetlana Tepteva, Pietro Zerbi, Laura Calvillo.

**Methodology:** Veronica Redaelli, Simonetta Papa, Gerardo Marsella, Laura Calvillo.

**Project administration:** Laura Calvillo.

**Resources:** Veronica Redaelli, Simonetta Papa, Gerardo Marsella, Giuliano Grignaschi, Nicola Ludwig, Fabio Luzi, Pietro Veglianese, Silvia Mazzola, Pietro Zerbi, John V. Roughan, Gianfranco Parati, Laura Calvillo.

**Software:** Veronica Redaelli.

**Supervision:** Laura Calvillo.

**Validation:** Veronica Redaelli, Simonetta Papa, Laura Calvillo.

**Visualization:** Veronica Redaelli, Simonetta Papa, Alice Bosi, Laura Calvillo.

**Writing – original draft:** Laura Calvillo.

**Writing – review & editing:** Veronica Redaelli, Simonetta Papa, Gerardo Marsella, Giuliano Grignaschi, Alice Bosi, Nicola Ludwig, Fabio Luzi, Irma Vismara, Pietro Veglianese, Silvia Mazzola, Pietro Zerbi, Luca Porcu, John V. Roughan, Gianfranco Parati, Laura Calvillo.

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
