## [Decision Letter · Decision Letter 0]

2 Sep 2019

PONE-D-19-20984

A Refinement approach in a mouse model of rehabilitation research.

PLOS ONE

Dear PhD Calvillo*,

Thank you for submitting your manuscript to PLOS ONE. After careful consideration, we feel that it has merit but does not fully meet PLOS ONE’s publication criteria as it currently stands. Therefore, we invite you to submit a revised version of the manuscript that addresses the points raised by the reviewer during the review process.

We would appreciate receiving your revised manuscript by Oct 17 2019 11:59PM. To enhance the reproducibility of your results, we recommend that if applicable you deposit your laboratory protocols in protocols.io, where a protocol can be assigned its own identifier (DOI) such that it can be cited independently in the future. For instructions see: http://journals.plos.org/plosone/s/submission-guidelines#loc-laboratory-protocols

We look forward to receiving your revised manuscript.

Kind regards,

Chang-Qing Gao

Academic Editor

PLOS ONE

Journal Requirements:

1. Please amend either the title on the online submission form (via Edit Submission) or the title in the manuscript so that they are identical.

Reviewers' comments:

Reviewer's Responses to Questions

**Comments to the Author**

1. Is the manuscript technically sound, and do the data support the conclusions?

Reviewer #1: Yes

2. Has the statistical analysis been performed appropriately and rigorously? 

Reviewer #1: Yes

3. Have the authors made all data underlying the findings in their manuscript fully available?

Reviewer #1: No

4. Is the manuscript presented in an intelligible fashion and written in standard English?

Reviewer #1: Yes

5. Review Comments to the Author

Reviewer #1: 1) I do not see where data from experiments are being made available in supplementary information.

2) Additional information needed under "Husbandry" section": age and sex of mice studied (if only male, need to justify why sex was not considered as a variable), single or co-housed, pathogen status

3) Experimental designs between Step1 and Step2 are not perfectly comparable: timing of buprenophrine admin was 15 min before procedure in Step1, 15 after procedure in Step2...all are conditions same (except opioid treatment as variable). Need to explain and justify why this difference in admin times for buprenorphine between experiments.

4) Video-recordings are a good supplement to assessment of pain, but other observations, such as heart rate, respiratory rate, vocalization, which all are altered in response to pain, cannot be assessed by videorecording. Therefore, authors should explain and justify why these primary observations were not part of the study, and/or why they feel that they are insufficient alone or can be replaced by observations made by videorecording.

5) Experimental design: there is no non-surgical control, which I can understand since the intention of the experiment was not to assess they surgery, but to assess two different pain relief protocols after surgery.

6) I do not understand how this is a double-blinded study. The observers were blind to the treatment, which makes this a single blinded study. But what other parameter were the observers blinded to in order to determine that this was double blinded?

7) Please indicate who is the authorizing "authority" for the method of euthanasia.

8) Twitching can be considered a spinal reflex that does not require conscious perception. Therefore, the authors should address whether carprofen could impact a spinal reflex directly without having an impact on conscious perception of pain. On the other hand, the grimace test is a far more direct indicator of conscious pain which was shown experimentally to be impacted by opioid treatment.

9) Temperature values were reported different between the two groups of mice receiving buprenorphine in Step1 and Step2 experiments. If this difference was spurious, why are not the other differences (between buprenorphine alone and with carprofen) also spurious?

10) Figures are of very low quality and very difficult to read or understand.

6. PLOS authors have the option to publish the peer review history of their article (what does this mean?). If published, this will include your full peer review and any attached files.

Reviewer #1: No

---

## [Author Response · Author response to Decision Letter 0]

8 Oct 2019

1. Is the manuscript technically sound, and do the data support the conclusions?

Reviewer #1: Yes

We thank the Reviewer for this recognition

2. Has the statistical analysis been performed appropriately and rigorously? 

Reviewer #1: Yes

Again, thank you for the acknowledgement

3. Have the authors made all data underlying the findings in their manuscript fully available?

Reviewer #1: No

We thank the Reviewer for bringing this to our attention A Supplement file with supporting information is now available. A complete raw data file is now provided.

4. Is the manuscript presented in an intelligible fashion and written in standard English?

Reviewer #1: Yes

Thank you for your approval

5. Review Comments to the Author

Reviewer #1: 

1) I do not see where data from experiments are being made available in supplementary information.

Thank you for bringing this inaccuracy to our attention. A Supplement file with supporting information is now available, in particular the complete raw data file is now provided.

2) Additional information needed under "Husbandry" section": age and sex of mice studied (if only male, need to justify why sex was not considered as a variable), single or co-housed, pathogen status.

Again thank you for the comment. We used males mice (averaged weight 25 gr, which correspond to about 2 months of age) to avoid possible hormonal interferences, thus preventing potential bias in our results. Two mice for cage were housed after surgery and animal facility is classified as SPF (specific pathogen free). These data are now included in the manuscript

3) Experimental designs between Step1 and Step2 are not perfectly comparable: timing of buprenophrine admin was 15 min before procedure in Step1, 15 after procedure in Step2...all are conditions same (except opioid treatment as variable). Need to explain and justify why this difference in admin times for buprenorphine between experiments.

Thank you for your note. STEP-1 timing of BUP administration represents the most common analgesic treatment for Spinal Cord Injury Model and, being administered before surgery, ensures the best possible analgesic effectiveness. One of our aims was to find an even more effective analgesic strategy with fewer side effects, therefore we wanted to combine the advantage of having analgesic efficacy due to pre surgical administration, with the advantage of not risking a possible respiratory depression, always possible when administering together anesthetics and opioids. Therefore, in step 2 we decided to perform carprofen pre-treatment to maintain analgesic effectiveness and to use buprenorphine right after surgery to avoid too much stress with two injections for the awaken animals before surgery. Moreover, in this way we avoided possible buprenorphine related respiratory depression during anaesthesia. 

4) Video-recordings are a good supplement to assessment of pain, but other observations, such as heart rate, respiratory rate, vocalization, which all are altered in response to pain, cannot be assessed by videorecording. Therefore, authors should explain and justify why these primary observations were not part of the study, and/or why they feel that they are insufficient alone or can be replaced by observations made by videorecording.

The Reviewer comment is very appropriate and highlights the issue of assessing pain in mice, which is one of the main goals of our group. Regarding heart rate and respiratory rate, according to our approach (which is not to interfere with main study endpoint) we did not measure the cardiovascular parameters. In fact, the main tools available for cardiovascular measurements are telemetry, ECGenie System or subcutaneous needle connected to a polygraph. The first was not possible because the device might affect locomotor parameters, in the second system, the drag of the legs due to the paralysis can interfere with the signal. The last procedure was again not possible since it would require anaesthesia, thus undermining the measurement. With regard to vocalization, mice are typically hunted animals and, respect to the rats, they more often neglect to show any vulnerability including vocalization. 

5) Experimental design: there is no non-surgical control, which I can understand since the intention of the experiment was not to assess they surgery, but to assess two different pain relief protocols after surgery.

The comment is absolutely pertinent, nevertheless, in Spinal Cord Injury, the main endpoint is the evaluation of Basso Mouse Scale for locomotion assessment. This parameter is not affected by sham procedure, therefore, in order to Reduce the number of animals and to avoid unnecessary suffering, we decided to exclude this group.

6) I do not understand how this is a double-blinded study. The observers were blind to the treatment, which makes this a single blinded study. But what other parameter were the observers blinded to in order to determine that this was double blinded?

We thank the Reviewer for reporting this error, this is a single-blinded study. The text has been corrected accordingly

7) Please indicate who is the authorizing "authority" for the method of euthanasia.

The method of euthanasia was approved by Italian Ministry of Health (Permit Number 62/2014-PR) according to 26/2014 Italian Law on the protection of animals used for scientific purposes.

8) Twitching can be considered a spinal reflex that does not require conscious perception. Therefore, the authors should address whether carprofen could impact a spinal reflex directly without having an impact on conscious perception of pain. On the other hand, the grimace test is a far more direct indicator of conscious pain which was shown experimentally to be impacted by opioid treatment.

We thank the Reviewer for this important comment. The first description of twitching behaviour in the context of pain detection was by one of our co-authors (Roughan; Pain 2001, 90: 65-74). This activity was originally defined as “a transient, apparently involuntary muscle spasm”, it is not a true reflex. We agree that reflexive responses are probably less helpful in terms of pain evaluation than scales such as the MGS, but reflexes can be defined as “an action that is performed without conscious thought as a response to a stimulus”. Pain is a stimulus but is more appropriately defined as a state. In Roughan’s article twitching ‘behaviour’ occurred in rats in response to abdominal pain with a proportionately similar level of occurrence as other more complex behaviours such as back arching and writhing, but was absent in intact rats given a range of dose rates of carprofen. Twitching is more frequent and easier to recognise, consequently it can be provide a useful insight as to the underlying pain state and whether it has been ameliorated by analgesic treatment(s). It is therefore a helpful parameter for pain assessment, especially when combined other parameters as we have done here.

9) Temperature values were reported different between the two groups of mice receiving buprenorphine in Step1 and Step2 experiments. If this difference was spurious, why are not the other differences (between buprenorphine alone and with carprofen) also spurious?

We thank the Reviewer for this relevant comment which allows us to clarify an important point: Infrared Thermography is a particular technique which is influenced by several parameters including

- variations in microenvironment temperature

-possible fast over activity, typical of small rodents, occurred some minutes before the measurement

-normal and accepted variations in Animal Facility temperature (22±2°C, according to Italian Law Dlgs. 26/2014)

Therefore, we did not compare baselines from STEP-1 with those from STEP-2, but we completely repeated the measurement in order to have a better internal control (not only for IRT values but also for the other measurements, including those related to buprenorphine w/out carprofen analgesic effects). Accordingly, STEP-1 results indicated mice temperature trend respect to time. STEP-2 indicated both the trend and the difference between analgesic treatments. In conclusion, with this design we obtained two information: a description of mice thermoregulatory response after SCI surgery and a possible correlation between thermography and pain.

10) Figures are of very low quality and very difficult to read or understand.

We apologize for the inconvenience which is probably due to a problem generated by the system in the website, when converting tiff file to pdf file for revision. We asked Editors for support and they answered:

Reviewer can access high-resolution versions of each image in the PDF by clicking the blue link at the top of the page where the figure appears.

We verified that the link actually opens the Tiff figures. We hope that this will fix the problem

6. PLOS authors have the option to publish the peer review history of their article (what does this mean?). If published, this will include your full peer review and any attached files.

Do you want your identity to be public for this peer review? For information about this choice, including consent withdrawal, please see our Privacy Policy.

Reviewer #1: No

---

## [Decision Letter · Decision Letter 1]

11 Oct 2019

A Refinement approach in a mouse model of rehabilitation research.

PONE-D-19-20984R1

Dear Dr. Calvillo*,

We are pleased to inform you that your manuscript has been judged scientifically suitable for publication and will be formally accepted for publication once it complies with all outstanding technical requirements.

With kind regards,

Chang-Qing Gao

Academic Editor

PLOS ONE

Additional Editor Comments (optional):

Reviewers' comments:

Reviewer's Responses to Questions

**Comments to the Author**

1. If the authors have adequately addressed your comments raised in a previous round of review and you feel that this manuscript is now acceptable for publication, you may indicate that here to bypass the “Comments to the Author” section, enter your conflict of interest statement in the “Confidential to Editor” section, and submit your "Accept" recommendation.

Reviewer #1: All comments have been addressed

2. Is the manuscript technically sound, and do the data support the conclusions?

Reviewer #1: (No Response)

3. Has the statistical analysis been performed appropriately and rigorously? 

Reviewer #1: (No Response)

4. Have the authors made all data underlying the findings in their manuscript fully available?

Reviewer #1: (No Response)

5. Is the manuscript presented in an intelligible fashion and written in standard English?

Reviewer #1: (No Response)

6. Review Comments to the Author

Reviewer #1: (No Response)

7. PLOS authors have the option to publish the peer review history of their article (what does this mean?). If published, this will include your full peer review and any attached files.

Reviewer #1: No

---

## [Editor Report · Acceptance letter]

21 Oct 2019

PONE-D-19-20984R1 

A Refinement approach in a mouse model of rehabilitation research. Analgesia strategy, Reduction approach and infrared thermography in spinal cord injury

Dear Dr. Calvillo:

I am pleased to inform you that your manuscript has been deemed suitable for publication in PLOS ONE. Congratulations! Your manuscript is now with our production department. 

With kind regards,

on behalf of

Dr. Chang-Qing Gao 

Academic Editor

PLOS ONE